# Pulmonary Drug Delivery of Antimicrobials and Anticancer Drugs Using Solid Dispersions

**DOI:** 10.3390/pharmaceutics13071056

**Published:** 2021-07-10

**Authors:** Hisham Al-Obaidi, Amy Granger, Thomas Hibbard, Sefinat Opesanwo

**Affiliations:** The School of Pharmacy, University of Reading, Reading RG6 6AD, UK; a.granger@student.reading.ac.uk (A.G.); t.hibbard@pgr.reading.ac.uk (T.H.); s.t.opesanwo@student.reading.ac.uk (S.O.)

**Keywords:** cocrystals, coamorphous, solubility, dry powder inhalers, pulmonary drug delivery

## Abstract

It is well established that currently available inhaled drug formulations are associated with extremely low lung deposition. Currently available technologies alleviate this low deposition problem via mixing the drug with inert larger particles, such as lactose monohydrate. Those inert particles are retained in the inhalation device or impacted in the throat and swallowed, allowing the smaller drug particles to continue their journey towards the lungs. While this seems like a practical approach, in some formulations, the ratio between the carrier to drug particles can be as much as 30 to 1. This limitation becomes more critical when treating lung conditions that inherently require large doses of the drug, such as antibiotics and antivirals that treat lung infections and anticancer drugs. The focus of this review article is to review the recent advancements in carrier free technologies that are based on coamorphous solid dispersions and cocrystals that can improve flow properties, and help with delivering larger doses of the drug to the lungs.

## 1. Introduction

The respiratory tract is susceptible to a range of conditions, such as viral, bacterial and fungal infections [1], which in turn can result in an exacerbation of other existing conditions through inflammation [2]. Lower respiratory tract infections which include bronchitis, tuberculosis and pneumonia, are classed as two of the leading causes of death, while pneumonia is a leading cause of death in children globally [3]. While inflammation can affect the entire respiratory system, different pathogens will inhabit different parts of the respiratory tract. For example, tuberculosis causative microorganism *Mycobacterium tuberculosis* colonises the lungs, deep within, and also inside the lung’s alveolar surfaces [4]. In cystic fibrosis (CF) patients, the bacterium, *Pseudomonas aeruginosa*, inhabits the conducting and respiratory zones of the lungs and is associated with recurrent infections. This is mainly caused by bacterial transformation to the biofilm producing mucoid strain, which exhibits increased resistance to both antibiotics and natural lung defence mechanisms, such as phagocytosis.

In addition to respiratory tract infections, lung cancer is a major cause of death. It is estimated that 1.6 million people die every year from lung cancer, making it one of the most fatal cancers [5]. Lung cancer remains difficult to cure using chemotherapy, as evident by the low long-term survival rate of patients [6]. Apart from the frequent low efficacy, cancer treatments have been associated with significant side effects. A study that included 449 cancer patients, revealed 86% of patients experienced at least one side effect from cancer chemotherapy, with 67% having experienced six or more side effects [7]. This calls for the exploration of new approaches, aimed at reducing the side effects, and in turn, improving patient tolerance [8], and further improve the quality of care provided to patients.

The pulmonary route has been used to treat different lung conditions, such as asthma and chronic obstructive pulmonary disease (COPD). Drugs delivered to the lungs are typically formulated in low doses; however, there is an increasing clinical need to deliver higher doses. For example, when antimicrobials are delivered directly to the lungs, higher doses ensure optimum lung concentration for tackling the infection [9]. Hence, a major challenge is to formulate these drugs with efficient deposition, while maximising the delivered dose. Current technologies are based on formulating the required doses of the active pharmaceutical ingredient (API) with the addition of excipients (known as carrier particles) [10]. These carrier particles serve to minimise undesired deposition into the oropharyngeal region, as well as reducing loss in the inhalation device itself. Among approaches to deliver larger doses of the drug to the lungs is solid dispersions. These are molecular mixtures of the drug and a miscible carrier by which properties can be tailored to achieve optimum outcomes, such as improved solubility [11]. The carrier can be a small molecule known as a coformer or a polymeric carrier leading to the formation of crystalline or amorphous dispersions.

Although oral formulations have a more enhanced therapeutic profile when it comes to treating systemic diseases, inhaled drug formulations are better at targeting lung conditions, such as infections and cancer. The aim of this review article is to explore the respiratory route for drug delivery, highlighting its advantages and challenges to deliver larger doses. While there have been different approaches to maximise the dose that can be delivered, the focus of this review is on recent advancements in the delivery of particulates prepared as solid dispersions. The use of solid dispersions represents a novel approach to deliver larger amounts of the drug while maintaining enhanced physicochemical properties. These physical molecular complexes (i.e., solid dispersions) can be engineered to modify properties, such as adhesion, aerodynamic diameter and morphology, allowing enhanced pulmonary drug delivery.

### Physiology of the Lungs and Factors Affecting Particles Deposition

The pulmonary alveoli exhibit a large surface area of over 100 m^2^ and thin walls of less than 1 μm, allowing fast absorption of drugs into the rich blood supply for systemic effect [12]. This would be beneficial for a number of drugs as the fast absorption would result in a more rapid onset of action compared to other administration routes, such as oral administration. As well as being advantageous for drugs acting both locally and systemically, it also has reduced systemic side effects that are common with other administration routes [13]. It is considered a non-invasive form of drug administration and often require lower doses compared with other systemic drug delivery routes [14]. The local administration also avoids first pass liver metabolism, which is detrimental for some drugs. Thus, the pulmonary route may be favoured over other parenteral routes, or when absorption via the gastrointestinal (GI) tract is inappropriate or ineffective [12,15].

The respiratory system comprise 23 generations (G0–G23) which have varying sizes, structures and functions [16]. Each generation splits into two smaller daughter branches to give the next generation. Altogether, these generations are divided into two zones: Conducting and respiratory. The conducting zone includes the structures from the trachea to the bronchioles from G0 to G16. The main role of the conducting zone is to carry the air into the lungs. The respiratory zone includes structures from the respiratory bronchioles to the alveolar ducts and alveoli from G17 to G23. This zone contains functional tissues where gaseous exchange occurs [17].

As a result of the varying size and structure of the lungs, inhaled particle size affects how deep into the lungs the drug and excipients can penetrate. For example, particles with a mass median aerodynamic diameter (MMAD) of 10 μm or larger tend to be deposited in the oropharyngeal region, whereas smaller particles with an MMAD of less than 3 μm can penetrate much deeper through the lungs into the alveoli [18]. MMAD is a measurement used to define the size of aerosol particles. The aerodynamic diameter of a particle relates to a sphere with the same density as water (1 g/cm^3^) which settles at the same velocity as the particle of interest in still air [19]. Using the mass median value, the aerosol size distribution is divided in half [20].

Particles intended for pulmonary drug delivery are categorised according to their size. The ideal particle size for inhalation is said to be <5 µm [21], with <3 µm having an 80% chance of reaching the lower airways and 50–60% chance of reaching the alveolar regions [22]. Particles > 5 μm are identified as coarse particles; fine particles range from 0.1–5 μm, and ultrafine particles are <0.1 µm. A monodisperse aerosol is said to be highly desirable for maximum deposition and specific targeting in the lungs [23]. However, there have been reports showing minimum differences in deposition between polydisperse and monodisperse aerosols [24]. The deposition of inhaled particles within the regions of the lungs is dependent on a number of physiological and pharmaceutical factors, such as particle shape, particle size, surface morphology, the breathing rate of the patient, lung volume and health condition of the patient. The physicochemical factors that affect particles deposition in the respiratory system are shown in Table 1. Particle deposition can occur through the following mechanisms: sedimentation, impaction and diffusion. Diffusion is a fundamental mechanism of particle deposition for particles < 0.5 μm. The process is influenced by Brownian motion—in other words, motion increases with decreasing particle size and particles move from a higher concentration to a lower concentration leading to the deposition upon contact with the airway walls. This mechanism heavily influences deposition in the lower regions of the lungs and the alveoli. Gravitational sedimentation can happen at a later region, typically within the tracheobronchial region (approximately the last six generations), as a result of the relatively low air velocity within this region [25]. In fact, as the residence time is longer, a combination of both sedimentation and free diffusion can occur. It is vital for drug absorption to occur, that the particles are deposited before exhalation takes place. Typical size depends on the aerodynamic diameter, which is approximately above 0.5 µm for sedimentation and below that for diffusion [26,27]. The bigger the particle and the lower the airflow rate, the faster the sedimentation in which inertial impaction plays a significant role for particles > 5 μm [28].

## 2. The Necessity to Deliver Larger Doses to Treat Lung Infections and Cancer

There are a variety of inhaled products for the treatment of conditions affecting the respiratory system (Table 2). As can be seen in Table 2, the maximum dose that has been successfully formulated for inhalation, thus far, is colistimethate sodium, with doses ranging from 80 to 125 mg, formulated as inhalation powder. Many of the higher inhalable doses are administered via the use of nebulisers, which may require guidance and can also be complicated for patients to use [32]. Desgrouas and Ehrmann reviewed the available evidence and called for developing inhaled antibiotics, especially for mechanically ventilated patients [33]. However, the majority of inhalable drug formulations are available in the dose range of micrograms and are indicated for conditions, such as asthma and COPD. In addition, there is a limited number of inhaled medications available as combinations, which are again mostly limited for use in asthma and COPD.

There are two major lung conditions that require high doses of medications, these being infections and malignancy. For lung infections, the causative bacteria and viruses can be found throughout the different structures of the lungs, which will alter the efficacy of treatment. The conducting zone of the respiratory system, which consists of the trachea, bronchi and bronchioles, plays a role in trapping microbes in mucus produced by goblet cells. The mucus is then transported by ciliated cells located on the epithelium to the oropharynx, where the mucus is either swallowed or removed via coughing [43]. However, in cases where the microbes go on to develop into infections, it is important to determine the specific site in the lungs where the infection has developed to effectively target and deliver the drug directly to the causative microbes. Table 3 lists common loci of prevalent microorganisms within the respiratory tract noting that several pathogens found in the respiratory tract are associated with the development of biofilms.

Upper respiratory tract infections typically occur in the conducting zone, examples of which include common cold, sinusitis and pharyngitis, whereas lower respiratory tract infections happen in the respiratory zone and include pneumonia and bronchitis. In addition, some microorganisms, such as *Haemophilus influenzae*, that are normally found in the conducting zone can go on to migrate to the respiratory zone, subsequently causing lower respiratory tract infections [44]. Some studies suggest that the conducting zone of the respiratory tract is the more common zone where bacteria and viruses are contained during infections. One study found that for *Pseudomonas aeruginosa* biofilms in cystic fibrosis patients, the majority of the bacteria were present in the conducting zone, which acted as a reservoir for the bacteria to multiply and form biofilms [45]. Consequently, it is important for inhaled formulations of antibacterials and antivirals to reach and deposit in the conducting zone of the respiratory tract to effectively treat the infections. With the ability to locally deliver antimicrobial drugs for treatment, there could also be a further advantage of reducing antimicrobial resistance. With oral and intravenous administration, there is a risk of an accumulation in infection-free sites, leading to the development of antimicrobial resistance. In addition, a number of antimicrobials, such as tobramycin, can be toxic when given in repeated high doses systemically, which can occur when antimicrobials are prescribed for recurrent infections [46]. Hence, utilising local drug delivery directly into the lungs could reduce the risk of toxicity.

Similar to infectious conditions, lung cancer is another serious condition that requires large doses of drugs for treatment. Current survival rates for all stages of lung cancer include 40% of patients surviving for one year or more after diagnosis, 15% of patients surviving for five years or more after diagnosis and only 10% of patients surviving for 10 years or more after diagnosis [50]. The nature of lung cancer suggests that it is more effectively treated by direct delivery to the lungs. Sardeli et al., suggested the need to use inhaled immunotherapy as opposed to intravenous administration to avoid systemic side effects and achieve a localised effect [51]. Hence, pulmonary drug delivery provides a novel opportunity to avoid unwanted drug distribution and could achieve maximum deposition of the drug at the site of action. As such, there is the potential for higher concentrations of the drugs to reach the lungs compared with oral and intravenous administration. Lung cancer cells are prone to rapidly developing resistance to anticancer drugs, so higher doses are given to combat this challenge [52]. When administered systemically, chemotherapy drugs can cause toxic side effects affecting healthy organs, especially when administering large doses, which may be prevented by local administration of the drugs [53]. By targeting the lungs directly, this also means there is potential for an accumulation of the drug to build up in the tumour cells, rather than in the kidneys, liver and spleen, which is observed with systemic drug use [54]. This has the potential to beneficially impact patients’ treatment and increase the likelihood of remission and survival.

## 3. Challenges Associated with Drug Delivery to the Lungs

Whilst pulmonary drug delivery has its advantages, to ensure advantageous deposition in the lungs, drug particles must be within the optimum size for lung deposition (Table 4 and Figure 1). If the particle size is relatively large, the deposition will occur in the larynx causing irritation, and if the particle size is relatively small, the particles will be immediately exhaled from the lungs and will not be deposited [55]. It is important to note that drug particles should be deposited in the lungs in a high enough proportion for the API to be absorbed and produce a therapeutic effect. Unsurprisingly, it has been shown that higher drug deposition in the lungs leads to enhanced clinical benefits [56]. As mentioned above, a common approach to improve the flow properties of powder for inhalation is to use carrier particles (such as lactose monohydrate). This is based on combining the drug with the larger lactose particles, which strip from the drug particles inside the device in response to the high velocity created by inhalation [57,58]. Any escaped lactose will deposit in the throat, leaving the drug particles to carry on with their journey towards the alveoli. Hence, the main challenge would be to deliver high doses of the API without significant loss in the oropharynx region. Another challenge is that inhalation devices do not have the capacity to accommodate high masses of powders [59].

As shown by the data collated and presented in Table 4, two different studies used a combination of beclomethasone dipropionate with formoterol labelled with technetium-99. The first dissolved the particles in hydrofluoroalkane to give an MMAD of 1.3 μm via pressurised metered-dose inhaler (pMDI) [60], the second delivered the solid particles with MMAD of 1.5 μm via a NEXThaler^®^ dry powder inhaler (DPI) [61]. The lung deposition for the pMDI was 34.08% (standard deviation (SD) = 9.3%), and the DPI had a lung deposition of 55.2% (SD = 3.7%). This shows that even though the same APIs were used, and the particle sizes were similar, the way that the API is formulated plays a significant role in the lung deposition of the particles. It is apparent that the type of the API has an effect by comparing the lung deposition of albuterol and beclomethasone dipropionate with formoterol in the pMDI considered previously [62]. They both have an MMAD of 1.5 μm, are labelled with technium-99 and are administered using an pMDI; however, the lung deposition of albuterol was 56.3% (SD = 9.2%), which is considerably higher than the beclomethasone, which has a lung deposition of 34.08% (SD = 9.3%). This suggests that API type and formulation must also be carefully considered when formulating drugs for pulmonary drug delivery.

Furthermore, Table 4 shows that within the same administration device, the preparation method of the particle has an impact on the deposition within the lungs. One study compared two suspensions of beclomethasone dipropionate labelled with technetium-99 in a metered-dose inhaler, by which the first formulation was dissolved in hydrofluoroalkane, and the other was dissolved in chlorofluorocarbon [63], it demonstrated that the different preparation methods led to a difference in both MMAD and lung deposition. Therefore, this suggests that the type of excipients used must be carefully considered for optimum lung deposition.

**Table 4 pharmaceutics-13-01056-t004:** MMAD, lung deposition and preparation method of different inhaled APIs.

API	MMAD (μm)	Lung Deposition (%)	Preparation of API	Ref
Formoterol	0.8	31 ± 11	Labelled with technetium-99 and dissolved in hydrofluoroalkane (HFA) in an pMDI	[64]
Beclomethasone dipropionate	0.9	53 ± 7	Labelled with technetium-99 and dissolved in HFA in an pMDI	[63]
Fluticasone propionate	2	12 ± 7	Labelled with technetium-99 and dissolved in chlorofluorocarbon (CFC) in an pMDI	
Beclomethasone dipropionate	3.5	4 ± 11		
Albuterol (salbutamol)	1.5	56.3 ± 9.2	Labelled with technetium-99 in an pMDI	[62]
	3	51 ± 8.9		
	6	46 ± 13.6		
Beclomethasone dipropionate and formoterol	1.3	34.08 ± 9.3	Labelled with technetium-99 and dissolved in HFA in a pMDI	[60]
Ciclesonide	1	52 ± 11	Labelled with technetium-99 and dissolved in HFA in a pMDI	[65]
Beclomethasone dipropionate and formoterol fumarate	1.5	55.2 ± 3.7	Labelled with technetium-99 in a NEXThaler^®^ DPI	[61]

## 4. Impact of Drug Delivery Devices on the Extent of Pulmonary Drug Delivery

There are currently three major categories of delivery devices that are used for inhalation therapy: pressurised metered-dose inhalers (pMDIs), nebulisers and dry powder inhalers (DPIs). pMDIs are widely available, portable inhalers designed to releases a specific dose of the drug, following depression of an actuator, which the patient then inhales in a single inspiration or via a spacer device. Studies have found that many patients have difficulty in using these devices, due to the required coordination of pressing down on the actuator and inhaling the resultant mist simultaneously [66]. If patients do not have the required coordination to self-administer their medication using a pMDI, some of the drug is deposited in the mouth and at the back of the throat. As a result, less of the API will be inhaled than intended, so the correct dose will not be administered. This could potentially result in substandard treatment, leading to exacerbation or worsening of a patient’s condition. pMDIs are formulated as a suspension or solution with one or more APIs along with a propellant. The propellant is used to generate the pressure required to form micron-scaled droplets for inhalation [67]. Furthermore, pMDIs deposit less of the drug into the lungs, due to the high particle exit velocity with the actuations [68]. This means that it is more difficult to formulate higher drug doses for use with this type of inhaler device.

DPIs are portable inhalers that operate by breath actuation, thus removing the need for coordination of inhalation with actuation required for pMDIs. Breath actuation involves the powder being released from the inhaler when the patient inhales with enough force, known as inspiratory flow. The inhalers are formulated to contain powder forms of drugs, and this solid-state gives the drug improved stability compared to pMDIs formulations [67]. The use of powders also allows for greater potential when formulating higher dose medications. Dry powder inhalers are available in three types: single-unit dose, multi-dose reservoir and multi-unit dose. The powder is stored in either capsules or sealed blisters that are broken during actuation. In general, patients tend to find that multi-dose DPIs are preferable to single-unit dose DPIs, due to the convenience of not having to replace the capsule or blisters with every use of the inhaler.

Nebulisers are less popular inhalation devices compared to the other two devices and are available in two forms. The most common nebuliser is the jet nebuliser, which works by passing a liquid formulation along with compressed air through a narrow tube into a wide chamber, which causes a reduction in pressure. This reduction in pressure then forms micron-sized droplets, which can then be inhaled. The alternative type of nebuliser is the ultrasonic nebuliser, which breaks down the solution of the drug into inhalable droplets by piezoelectric vibrations [68]. Both types of nebulisers require a face mask or mouthpiece, which are bulkier and less portable than the other two devices, and therefore, less convenient for frequent and repeated use [69]. The way nebulisers are formulated are unsuitable for some drugs, particularly for drugs that are unstable when in solution [70]. Nebulisers also take longer to administer the drug when compared to other drug delivery devices, due to the time taken for the drug to pass through the chamber into the mouthpiece or mask, and with inhalation over a prolonged period. The use of nebulisers can also result in the administration of less precise doses as there can be deposition of the API particles in the chamber and mouthpiece or face mask. In general, nebulisers tend to be used by patients who cannot use the other devices, due to difficulties with coordination associated with pMDIs and/or the inability to produce adequate inspiratory flow associated with DPIs. The main patient groups these affect are young children, elderly populations, patients with COPD and patients with CF [71].

There is a limited number of licensed inhaled antimicrobials that are only available for patients suffering from recurrent infections as in CF (Table 5). Other antibiotics, such as gentamicin and amikacin, can be nebulised for pulmonary drug delivery by using the solution for injection if required; however, these cases are for unlicensed use in CF patients only and not for use for respiratory infections [72]. As can be seen in Table 5, nebuliser solutions and inhalation powders can be used to formulate higher doses of drugs, and hence, these are the devices currently used on the market. Tobramycin, colistimethate sodium and zanamivir are all available as dry powders for inhalation. Tobramycin inhalation powder is produced via an oil-in-water emulsion-based spray-drying process [69], and each particle consists of an amorphous tobramycin sulphate, and a gel-phase phospholipid 1,2-distearoyl-sn-glycero-3-phosphocholine (DSPC). Tobramycin inhalation powder comes in a device known as a TOBI^®^ Podhaler^®^, which consists of the engineered dry powder, the hard capsule, which is the primary packaging of the powder, and the device to administer the powder [70]. Other excipients used in the capsules include calcium chloride, which is used to stabilise the emulsion droplets during the spray drying and sulphuric acid for pH adjustment [71,73]. Each inhalation delivers 28 mg of tobramycin from the capsule to the lungs; however, four capsules are required for each dose, and therefore, eight inhalations per day. Colistimethate sodium is an inactive prodrug of colistin (polymyxin E), and is formulated as such to reduce the toxicity, typically nephrotoxicity, that is often experienced by patients when taking colistin. It is formulated to be administered using a Turbospin^®^ inhaler, under the brand name Colobreathe^®^ [74]. The powder is micronised and filled in polyethylene glycol (PEG)-gelatin hard capsules, which also contain purified water and sodium lauryl sulphate as excipients [75,76]. Zanamivir is an antiviral drug used for the treatment and prevention of influenza. It is formulated under the brand name Relenza^®^ for its inhalation powder, and is produced via air-jet milling to contain 5 mg of zanamivir and 20 mg of lactose monohydrate particles as carrier particles in each double-foil blister [77].

## 5. Pulmonary Drug Delivery Using Carrier Free Technology

Large carrier particles, such as alpha-lactose monohydrate, are added to the API to form an inhaled powder with enhanced formulation properties. These properties include the flowability and aerosol dispersion of the powder to maximise the number of API particles reaching the site of action. Lactose is available in a variety of grades for inhalation, with median diameters ranging from <5 μm to 250 μm [82]. During formulation, micronised API particles with an MMAD of between 1 μm and 5 μm are attached to the surface of these much larger carrier particles. Physical interactions hold the API in place, which prevents agglomeration of micronised API, due to cohesive forces. On inhalation, API is detached from the carrier and aerosolised, due to force exerted by the inhaler design in the form of friction, inertia and drag [83]. By retaining micronised API particle size, maximum dispersion of powder into the lungs is seen, due to increased aerosolisation [84]. This influences the drug administration in addition to preventing problems during the manufacturing process of the powder [85].

Carrier-based DPIs are the most prevalent formulation of DPIs and tend to be used to deliver relatively higher doses of API compared to pMDIs. However, certain mass ratios of API to carrier particles must be maintained to prevent agglomeration, as for example these can vary depending on the type of the drug, ranging from 1:2 to 1:4 in some cases [86]. For example, Relenza^®^ requires a 25 mg powder to be inhaled so that to deliver 5 mg of zanamivir (i.e., a mass ratio of 1:5). These ratios create a maximum practical dose that can be delivered to the lungs through dry powder inhalation, making high dose delivery even more challenging. Even with carrier particles enhancing aerosolisation, deposition in the lungs is still relatively low, and as a result, the amount of API to reach the site of action is low. Fine particle fraction (FPF) refers to the fraction of particles with a size smaller than the respirable size divided by the total emitted dose of the inhaled API [87]. DPIs that do not contain carrier particles, therefore, have greater potential for delivering higher doses to the lungs.

Carrier free DPIs are a new-generation system that use special excipients and technologies, such as crystal engineering, to formulate micronised API particles whilst addressing the limitations explained above. These methods can increase the amount of lung deposition, with some formulations reporting FPF values of 63% [88]. One of the benefits of developing carrier free DPI formulations is that there can be an increased mass of API without having to retain large masses of the carrier particles. Therefore, this can maximise the dose available in each inhalation through maximising the amount of API that is deposited in the lungs. There are several dry powder formulations that have been or are currently being investigated to produce carrier free dry powder inhalers, as shown below in Table 6. As can be seen, different doses have been tested with some formulations delivering doses up to 50 mg, as displayed in Figure 2.

Many different methods have been used to develop dry powders for inhalation, also shown in Table 6, and each method produces variation to yield, cost of production and manufacture time. This data shows the difference in the size of the doses able to be formulated in DPIs for inhalation, compared to current doses available for inhalation (Figure 2). This also suggests that the use of carrier free inhalation technology can play a key factor in addressing the challenge of increasing doses of drugs for pulmonary drug delivery. Also noted is the wide range of drug classes that are being considered with regards to delivery via dry powder inhalation, including non-steroidal anti-inflammatory drugs (NSAIDs, i.e., ibuprofen and indomethacin), antimicrobials (i.e., tobramycin and netilmicin), antihistamines (i.e., ketotifen), and phosphodiesterase type 5 inhibitors (i.e., sildenafil). It was also noted that for many antimicrobials, there is the possibility of creating drug combinations within the same device—a useful tool for patients who require complex antimicrobial treatment.

## 6. Pulmonary Drug Delivery on the Nanoscale

Nanoparticulate formulations maintain nanoscale particles of the drug through encapsulation within inhalable size particles to prevent aggregation and ensure deposition of nanoparticles in the lung. There are several particle engineering methods to achieve nanoparticle formulation, such as the use of liposomes, solid lipids and polymers, which have been reviewed elsewhere [103,104,105]. It is common for these formulations to be stored as suspensions then delivered to the lung through a nebuliser; however, carrier free, dry powder nanoparticle formulations are also being developed.

Zhu et al., described the formulation of poorly soluble ivacaftor (Iva) as bovine serum albumin nanoparticles, which were then spray freeze-dried with different ratios of soluble colistin (Co) matrix and L-leucine [106]. All subsequent formulations showed high emitted dose following inhalation for both APIs (>90%); however, a range of FPF values (%) were seen corresponding to initial solid content in the formulations. Dissolution rates of the best performing nanoformulation and a jet-milled physical mixture of Iva and Co were then compared. After 3 h, the nanoformulation showed three times greater dissolution of Iva compared to the physical mixture control, with the concentration of Iva dissolved equivalent to highly soluble Co. Enhanced dissolution of Iva was attributed to the albumin maintaining Iva nanoscale particle size and amorphous form. The use of Co as a water-soluble matrix both enhanced the dispersion of Iva nanoparticles, as well as broadening the formulation clinical scope for patients with cystic fibrosis.

Doxorubicin nanoparticles (DNPs) were formulated as a colloid through emulsion polymerisation then spray freeze-dried with lactose to form particles of inhalable size [107,108]. DNPs showed increased survival rates compared to IV doxorubicin controls (including IV DNPs) in a rat model and much lower cardiotoxicity compared to doxorubicin DPI control. DNPs were also formulated with sodium carbonate to enhance the release of doxorubicin from the formulation. When inhaled, the effervescent formulations showed increased survival rates compared to DNP controls showing the importance of an active release mechanism from the formulation. In a slightly different approach, often utilising nanoscale dimensions to enhance inhalation properties, the use of porous particles has been investigated and was shown to improve efficacy of inhalation [109,110,111].

## 7. The Design of Carrier Free Formulations Using Coamorphous Solid Dispersions (CACDs)

Most solid APIs exist in a crystalline state held together by strong intermolecular bonds, and therefore, display good stability profiles (Figure 3). However, the crystalline state often shows poor solubility, due to the high energy required to break the crystalline lattice which creates a major problem for developing new APIs [112,113]. There has been much interest in the process of ‘drug amorphisation’, to address poor solubility, which involves the conversion from a crystalline state to an amorphous solid state. The amorphous solid state offers an improved solubility and dissolution rates as a result of possessing higher entropy [114,115,116]. The advantages associated with higher energy forms are often negated, due to recrystallisation to a more thermodynamically stable form during processing and storage [117,118,119]. This can limit applications, and so the production and maintenance of amorphous drugs with adequate stability remain a challenge. Formulation strategies based on solid molecular dispersions are being explored, including polymeric amorphous solid dispersions (PASDs), and more recently, coamorphous solid dispersions (CASDs).

PASDs incorporate low API loading within a compatible polymer to maintain the solubility advantage of amorphous systems through the formation of strong intermolecular attractions [120]. Examples of PASDs applications are seen in the combination of paclitaxel and polyvinylpyrrolidone (PVP), ritonavir and PVP, ciprofloxacin and polyvinyl alcohol [121,122,123]. However, the main limitation of this approach is the low level of API loading within the formulation, requiring increased dosing. In addition, many polymeric carriers are hygroscopic, which can lead to API recrystallisation while in some cases polymers can be unsuccessful in maintaining a good stability profile [123]. The applications and limitations of PASDs have been highlighted in a number of reviews [124,125,126].

CASDs are a relatively new formulation approach through which the crystalline drug is amorphised thermally or mechanically and stabilised with a small molecular weight coformer. These systems are often stabilised through strong intermolecular hydrogen bonding and exhibit apparent stability, due to an increased glass transition temperature [124]. Glass transition temperature refers to the temperature at which below, the system exists in an unstable glassy state and above, the system exists in a rubbery state [125]. CASDs can be obtained through different preparation methods, such as freeze drying, quench cooling, milling and solvent evaporation methods [126,127]. Milling is an example of mechanical amorphisation where a direct mechanical impact on the drug causes a disturbance in the crystalline structure, forming an amorphous state. Quench cooling and solvent evaporation are examples of thermal amorphisation where a crystalline solid state is prevented from reforming, due to molecular level interactions. Here, the drug is either in a liquid state following melting or dissolved to form a solution then rapidly cooled, or solvent removed rapidly. CASDs are believed to be able to provide an increase in drug kinetic solubility thanks to the high energy of the amorphous state driving faster dissolution [128]. In addition, improved physical stability is offered through an increase in the glass transition temperature (T_g_) and the presence of intermolecular interactions between the API and coformer, as reported by several research papers [124,129]. CASDs are said to reduce hygroscopicity, as API hydrogen bonding regions are strongly bound to the coformer. Therefore, hydrogen bonding with water is less favourable, which helps to maintain stability and particle integrity.

CASDs provide the opportunity for many types of formulations, such as drug-excipient and drug-drug combinations, and many have been reported as being successful. An example of a drug-drug combination is the spray drying of colistin with azithromycin to treat infections caused by multi-drug resistant bacteria [130]. A study conducted by Wang et al., showed an enhanced solubility profile between two APIs with low aqueous solubility: lacidipine and spironolactone [131]. Many other drug-excipient combinations are listed in Table 7, by which commonly used coformers in CASDs for pulmonary drug delivery include mannitol, sugars and amino acids, such as leucine [132,133].

The spray drying technique is based on solvent evaporation which uses the processes of liquid atomisation, gas/droplet mixing and drying to create microparticles [134]. First, a liquid feed consisting of mixed components, typically pure API, excipient and a common solvent, is converted into smaller droplets via atomisation. Next, the droplets are sprayed downwards into a vertical drying chamber exposed to air or nitrogen at a temperature higher than the solvent boiling point [135]. Within this drying chamber, the droplets rapidly shrink in size to produce dry microparticles.

Through spray drying, aspects of particle morphology, such as shape, size and surface properties, can be highly controlled, which deems the technique appropriate for formulating inhalation powders. Many papers have controlled particle morphology through variations to the inlet temperature, the spray nozzle diameter and properties of the feed solution, such as concentration and viscosity [136]. The inlet temperature (also known as the drying temperature) and the feed rate are parameters of high importance as they control the rate of evaporation and drying load, which determine particle properties and product yield [137]. It is common to see outlet temperatures ranging from 70 °C to 105 °C with inlet temperatures ranging from 100 °C to 210 °C. A faster drying rate could lead to a higher product yield, as there are fewer particles adhering to the chamber walls, which may also implicate that solvent evaporation is not complete. The spray drying process is highly applicable for large scale manufacture, where the integrity of formulation is retained despite ‘upscaling’ [138]. Altogether, spray drying offers a great opportunity to incorporate a number of excipients into one formulation whilst improving the physical stability of the particles.

## 8. Examples of Coformers as Components of CAMs

L-leucine has been widely used in many studies to improve the aerosolisation properties of particles, due to its antihygroscopic effect and its ability to generate coarse particles, following surface enrichment. Momin et al., formulated spray dried kanamycin with various concentrations of L-leucine, showing the 5% L-leucine formulation to have the best improvement in aerosolisation properties [146]. Likewise, significant increases in aerosolisation properties were recorded for the other ratios of L-leucine. Other papers also reported L-leucine as a well-rounded excipient for improving the aerosolisation properties of powder particles. Interestingly Mangal et al., formulated spray dried azithromycin with various concentrations of L-leucine yet reported no significant change to powder morphology or FPF (*p* > 0.05) [132]. In addition, there was no significant increase seen in the percentage emitted dose (*p* > 0.05); however, an increased in vitro dissolution rate was reported. A lack of change to FPF and emitted dose can be explained when considering particle surface morphology and composition. It is believed that the more corrugated a particle surface is, the greater the resulting aerosolisation properties. This process is attributed to the success of L-leucine in improving aerosolisation. Chen et al., reported how changes to the composition of PASDs result in differences to the surface composition of particles, and therefore, flowability and aggregation [136]. Perhaps the surface enrichment of azithromycin with L-leucine for this formulation was poor, resulting in little change to particle surface morphology when compared to the control. When using L-leucine as a coformer, it is important that drying causes the distribution to the outer regions of droplets, causing recrystallisation of L-leucine on the particle surface, creating a corrugated effect [147]. This concept has also been documented by McShane et al. [138]. The impact of spray drying conditions on particles morphologies was recently highlighted in a different study by which spray dried coamorphous ciprofloxacin tartrate salt was shown to exhibit improved properties compared to ciprofloxacin alone [148].

The distribution of the components within the droplet during spray drying is said to be dependent on physicochemical properties of the components, such as solubility, the diffusivity of components and hydrophobicity [149]. These properties ultimately result in the final composition of the surface of spray dried particles. In the example mentioned above of azithromycin with L-leucine, the solubility of azithromycin is very low, and it likely had an impact on the distribution of L-leucine within the CASD. Li et al., conducted in vitro study to assess deposition, by which, spray drying azithromycin with mannitol resulted in low FPF values ranging between 38–42% [150]. Signifying the impact of preparation method and locality of excipients, Padhi et al., suggested that by increasing the ratio of L-leucine may improve surface enrichment, which would result in a substantial increase in the FPF [151]. In a different study, it was proposed that the presence of hydrophobic azithromycin helps to control humidity to prevent reduction in FPF [130].

Hassan et al., suggested that surface enrichment of spray dried materials is dependent on solvent evaporation rate and diffusion coefficients of the solutes [152]. Based on computational fluid dynamics, it was shown that the maximum rate of solvent evaporation proportionally affected MMAD [153]. It is believed the flow of the solute equates to that of the solvent; therefore, the movement of solute within a droplet is highly linked to solvent flow. Lower evaporation rates have a more significant impact on redistribution to droplet surface for solutes with high diffusion coefficients [149,152]. Shetty et al., when using water as a solvent, concluded that a low inlet temperature (<120 °C) was not considered, due to the possibility of it not being sufficient for drying [141]. Chen et al., also documented lower surface enrichment occurring, due to a reduction in the inlet temperature [136]. It is believed that the inlet temperature influences the particle shape, as well as surface enrichment. The inlet temperature indirectly affects the outlet temperature, and a lower outlet temperature is believed to result in a more spherical particle shape, which is detrimental to FPF [154,155].

Benke et al., reported spray dried meloxicam potassium (MXLspd) that was compared to the carrier-based meloxicam and lactose monohydrate InhaLac^®^ (µMXL + IH70) [21]. As expected MXLspd showed an improvement in FPF < 5 μm (59.47%) compared to µMXL + IH70 with an FPF < 5 µm (24.99%). In addition, µMXL + IH70 had an MMAD of 7.18 µm, therefore unfavourable for pulmonary drug delivery. When assessing the impact of humidity on FPF, authors observed a significant decline (*p* < 0.0001) observed in the FPF (<4.9 µm) between cospray dried ciprofloxacin with lactose when stored at 20% RH and 50% RH after 10 days. This may be due to the caking observed, which led to larger powder particles formed. Similar observations were reported by Guenette et al., which showed that larger lactose particles led to reduced flow properties [156]. The larger particle size may have occurred, due to particle agglomeration when stored at a higher RH. These examples highlight that for highly aerosolised powder, it is necessary to maintain the stability of spray dried CASDs when hygroscopic coformers are used. This is because of the presence of the excipient such as L-leucine, which has been reported to combat moisture-related stability issues, due to its hydrophobic nature [157]. Part of the reason why CASDs are successful is that the coformer changes from being a passive carrier to an essential constituent in the formulation [138].

In a separate study, Lababidi et al., further examined the success CASDs had in pulmonary drug delivery by which azithromycin and ciprofloxacin were spray dried with L-leucine and n-acetyl cysteine (NAC), an amino acid derivative and known mucolytic [129]. Mucolytics are used in combination with antibiotics to dissolve thick mucus found in the CF lung [157,158]. When mucolytics, such as NAC, are used as CASDs coformers, the formulation benefit becomes two-fold, creating a more effective treatment with better formulation characteristics. As mentioned before, the addition of L-leucine is aimed at reducing particle cohesion and increasing aerosolisation properties. However, in the study conducted by Lababidi et al., the main aim of adding NAC was to the formulation was to dissolve mucus, enhancing antibiotic effect against the bacteria. The authors showed that azithromycin/n-acetyl cysteine combination reduced *P. aeruginosa* biofilm by 25% at a concentration of 0.3 µm/mL compared to azithromycin and n-acetyl cysteine alone.

Forming amorphous dispersions composed of antimicrobials through complex formation with metal cations has been explored as a method for enhancing formulation characteristics of inhaled antimicrobials. Lamy et al., produced inhalable dry powder microparticles of ciprofloxacin, complexed with calcium or copper ions (Cip-Ca; Cip-Cu), through spray drying [159,160]. The resulting powders were amorphous, and both calcium and copper formulations showed similar shell-like morphology, characteristic of spray dried materials, and similar in vitro solution properties. When comparing pharmacokinetics in vivo, both formulations showed enhanced lung retention compared to the ciprofloxacin control, suggesting that forming metal counterion complexes can reduce the rate of transcellular diffusion from the lung fluid. It was also seen that the ratio of ciprofloxacin in epithelial lining fluid to plasma was five times greater for the Cip-Cu than Cip-Ca formulations. The authors linked this with the strength of complex which had formed, corresponding to: The enhanced stability of the Cip-Cu, a slower rate of absorption into plasma and longer elimination half-life. Thus, antibiotic lung retention can be controlled depending on the metal ion selected, leading to an optimised concentration in lung fluid.

The influence of metal salts on the inhalation properties of levofloxacin has also been analysed. Barazesh et al., cospray dried levofloxacin with four metal chloride salts with and without leucine [161]. Sodium (Na) and potassium (K) monovalent salts were compared with magnesium (Mg) and calcium (Ca) divalent salts. When measuring aerosol properties, most effects was seen for formulations containing the highest measured percentage of metal ion (i.e., 20% *w*/*w*). It was also observed that the addition of divalent salts reduced the FPF of formulations and caused higher water retention post spray drying. The authors concluded that the highly hygroscopic divalent metal salts are likely to uptake more water, leading to local crystallisation and particle aggregation. Monovalent salts at 20% *w*/*w* showed a higher FPF compared to lower concentrations. With respect to leucine formulations, FPF was significantly higher, corresponding to the characteristic corrugated effect seen when spray drying with leucine.

## 9. The Design of Carrier Free Formulations Using Cocrystals

The formulation of carrier free dry powders can be achieved by a variety of methods. These include the formation of salts, such as the generation of a sildenafil-citrate salt [92], forming complexes with cationic groups, such as the generation of indomethacin and polylysine microparticles [89], and forming CASDs, such as with ciprofloxacin and colistin [101]. Another route being investigated is the generation of cocrystals and seems to be a promising method of improving physicochemical properties if other methods are not suitable. For example, with the formation of salts, if the API is neutral, and therefore, does not contain an acidic or basic group that can be ionised, salts cannot be practically formed [162]. Cocrystallisation may also be seen as a more desirable method of physicochemical enhancement than others, for example, CASDs, due to enhanced physical and chemical stability [163].

Pharmaceutical cocrystals are multicomponent crystals formed between an API and pharmaceutically acceptable coformer. API and coformer exist in a fixed stoichiometric ratio held together through [164] non-covalent and non-ionic interactions. These interactions are intermolecular in nature and include hydrogen bonding, halogen bonding, π-π stacking and van der Waal forces [165]. Hydrogen bonds are considered the ‘key interaction’ for cocrystallisation, due to their strength, directionality and being commonly found in organic molecules [166]. Therefore, cocrystals are distinct from polymorphs, due to multiple components, and salts, due to lack of ionisation, and CASDs, due to their crystalline nature. The formulation of APIs as cocrystals is regarded as a physical modification to improve physicochemical properties, which maintains the API pharmacodynamic profile [167]. Since the cocrystal is a unique crystalline entity, it has unique physical properties, including solubility, dissolution, flowability and stability. In general, coformers are chosen based on molecular compatibility, as well as possession of desired physical property change. For example, to improve an API’s solubility, one would choose a coformer with compatible functional groups, as well as high solubility [168].

Cocrystallisation is currently attracting considerable interest, due to the improved physicochemical properties that can be introduced to the API [169,170]. For example, cocrystals of meloxicam–succinic acid were combined with PEG 4000 to enhance aqueous solubility [171]. Cocrystals have largely been studied with a focus on improving oral drug delivery, and it is only recently that studies have begun to explore how cocrystals can be used for other administration routes, including pulmonary drug delivery. It is important to focus on all changes to API physical properties rather than flowability and dispersion alone. For example, altering solubility and dissolution will influence retention and epithelium wall permeation which will determine either a local or systemic effect. Selecting the correct coformer can increase aerosolisation properties, as well as maintaining the balance between API retention in the lung and API clearance through various mechanisms [172].

M Karashima et al., produced micronised powder formulations of itraconazole for inhalation using cocrystallisation and jet-milling [163]. These cocrystal formulations showed superior aerodynamics compared with itraconazole control put through the same processing and amorphous itraconazole spray dried with mannitol corresponding with reduced particle size. Also tested were the intrinsic dissolution rates of cocrystal formulations compared to itraconazole and amorphous itraconazole in mock lung surfactant. After 1 h, the dissolved concentration of itraconazole from cocrystal formulations were between 5 and 10 times higher than itraconazole and the amorphous form. The authors reported higher plasma concentrations (C_max_) compared to itraconazole and the amorphous form of itraconazole. Comparing this finding with intrinsic dissolution data, supports the conclusion that the extent of dissolution in the lungs influences the extent of systemic partitioning.

Alhalaweh et al., produced several inhalable powders of theophylline through cocrystallisation with nicotinamide, urea and saccharin [173]. These were compared against theophylline control, but also against cocrystals formulated with lactose carrier. The authors comment that cocrystal formulations showed more favourable aerosolisation with less API loss, due to impaction. However, not all cocrystal formulations were better aerodynamically compared to the control. This highlights the unpredictable nature of coformer selection in cocrystal formation and the importance of considering all aspects of physical property changes. Tanaka et al., produced a theophylline:oxalic acid cocrystal through a combination of spray and freeze drying [174]. When compared to the control, the resulting formulation showed resistance to hygroscopicity and polymorphic transition whilst maintaining good pulmonary delivery. The authors attribute this stability to the low recorded energy state and close intermolecular interactions within the cocrystal.

API solution properties are a key component to determining API destination following inhalation. One study investigated how the properties of 5-fluorouracil differ from three different cocrystals created with gentisic acid (FUGA), 4,5 dihydroxybenzoic acid (FUBA) and 4-aminopyridine (FUPN). This study determined that the intrinsic dissolution of all three cocrystals were increased compared to pure 5-fluorouracil in both pHs of 1.2 and 6.8, with the largest difference being between 5-fluoruracil and FUPN. 5-fluorouracil had an intrinsic dissolution rate (IDR) of 0.12 mg cm^−2^/s in pH 1.2 and 0.18 cm^−2^/s in pH 6.8, whereas FUPN had an IDR of 0.7 mg cm^−2^/s in pH 1.2 and 0.92 mg cm^−2^/s in pH 6.8, showing more rapid dissolution from the cocrystal forms of 5-fluoruracil than the pure API [175]. Increased dissolution has also been observed in three diacerein cocrystals with isonicotinamide, nicotinamide and theophylline (DA-ISO), (DA-NIC) and (DA-THE). Diacerein had an IDR of 0.065 mg cm^−2^/min compared to 0.216 mg cm^−2^/min, 0.284 mg cm^−2^/min and 0.137 mg cm^−2^/min for DA-ISO, DA-NIC and DA-THE, respectively. This again shows that each cocrystal had a higher IDR value than the pure API molecule, suggesting a faster rate of dissolution [176].

Another study demonstrated the effects that cocrystals have on the pharmacokinetic profile of a drug. Daidzein and three cocrystals, with isonicotinamide (DIS), cytosine (DCYT) and theobromine (DTB) as the coformers, were compared. It was found that all three cocrystals had higher Cmax values, than daidzein (870.5 ng/mL), with DIS possessing the highest Cmax (1848.7 ng/mL). In addition, the time taken to reach the Cmax (Tmax) values for each cocrystal was 3 h compared to 4 h for pure daidzein [177]. Table 8 summarises a variety of cocrystal formulations, the method of manufacture used and the difference between API and coformer molecular weight. Shown is that cocrystallisation applied to a great variety of different APIs produced through a wide range of different production methods. Of note are the studies which use methods applicable for producing inhalable dry powders, such as spray drying, spray freeze drying and jet milling. It also gives an example of the wide variety of successful coformers available for cocrystallisation, suggesting a great variety in physical property variations even for the same API. Examples of drug:drug cocrystals are listed to show the possibility of creating synergistic therapies through a single phase crystalline powder, i.e., cocrystals. As shown in Table 8 and Figure 4, all coformers have molecular weights lower than alpha-lactose monohydrate (342.3 g/mol) and either much smaller or similar molecular weight to the API. This demonstrates how formulations containing coformers take up less volume in comparison to carrier-based delivery systems. As discussed earlier, the volume which lactose particles take up in formulations restricts the quantity of API that can be practically delivered. By avoiding the use of carrier particles, through coformer based methods, greater amounts of API particles can be formulated within a DPI device, possibly increasing the dose of API per actuation.

## 10. Conclusions

Coamorphous solid dispersions and cocrystals are attractive physical structures that have been used to improve the physicochemical properties of inhaled APIs to target infections and lung cancer. When developing pharmaceutical cocrystals/coamorphous for inhalation, careful selection of the coformer should be followed to improve the efficacy to deliver drugs to the lungs. It is likely that by creating cocrystals/coamorphous dispersions, the need for carrier particles in DPIs can be eliminated, and therefore, a greater amount of the API can be delivered. This would result in an increased dose in each inhalation, allowing lower potency drugs that require higher doses to be formulated for inhalation. It is important to understand the complexities of pulmonary drug delivery when considering the formulation of drugs for inhalation to ensure the drug particles have adequate aerosolization properties. This is necessary to ensure the particles can reach deep into the lungs at the site of action, without being aggregated or immediately exhaled from the lungs. It is important to consider all steps throughout drug development process, including the excipients that are used and the devices to administer the drug, because some of the devices have shown clear advantages when it comes to overcoming the challenges with formulating inhaled APIs. The complexity of diseases imposes greater challenges demanding more studies to optimise particles properties for better deposition. Ultimately, it is crucial to match the properties of the particles to the desired site of action, such as the case when targeting bacterial biofilms and cancer.

## Figures and Tables

**Figure 1 pharmaceutics-13-01056-f001:**
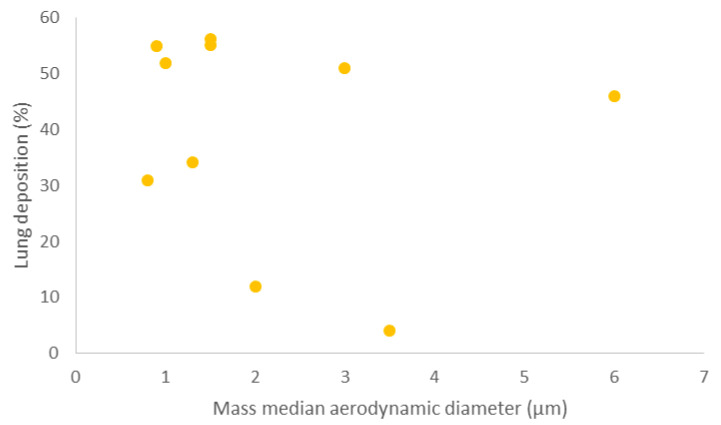
A scatter graph, comparing MMAD with corresponding lung deposition values in Table 4.

**Figure 2 pharmaceutics-13-01056-f002:**
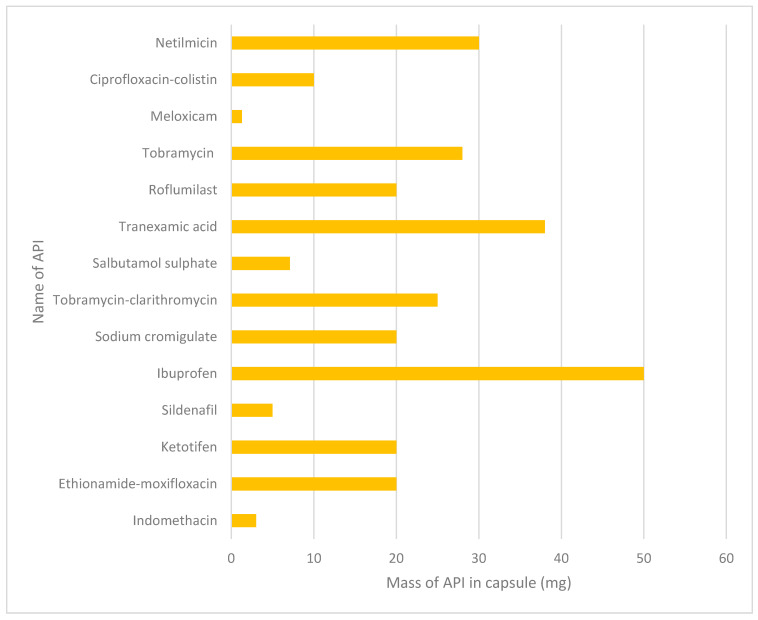
Bar graph for the weight of API in the capsule for each type of drug extracted from Table 6.

**Figure 3 pharmaceutics-13-01056-f003:**
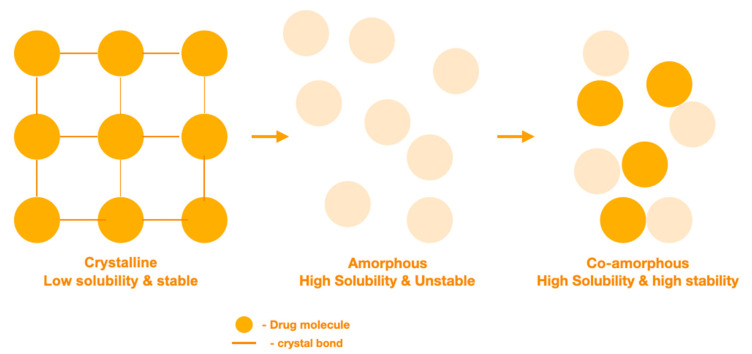
Illustration of the crystalline state, amorphous state and coamorphous state and their characteristics.

**Figure 4 pharmaceutics-13-01056-f004:**
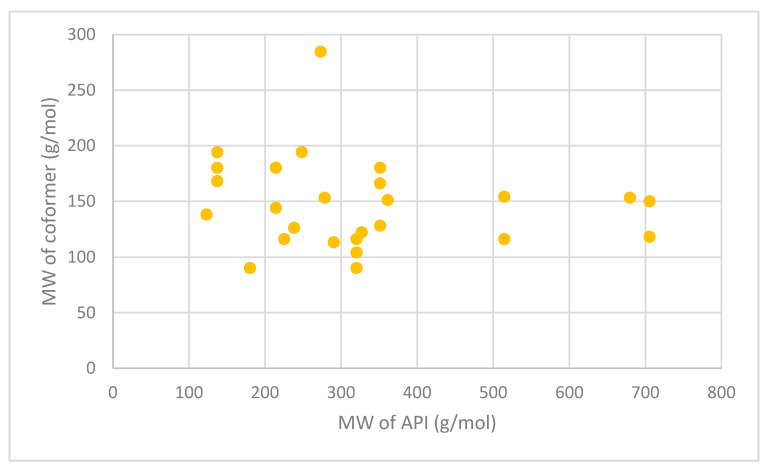
A scatter graph of the data from Table 8 comparing the molecular weights of the APIs and their coformers.

**Table 1 pharmaceutics-13-01056-t001:** Physicochemical factors to be considered when designing inhaled formulations. Each parameter influences both aerosolisation and deposition. Adapted from the work by the authors of [29,30,31].

Property Type	Parameter
Aerosol	Air/Particle velocity Mass median aerodynamic diameter Fine particle fraction
Particle	Bulk density Tap density Shape Charge Surface energy * Surface texture * Surface composition *
Physicochemical	Solubility Hygroscopicity

Note—* Factors specifically affecting aerosolisation.

**Table 2 pharmaceutics-13-01056-t002:** Examples of currently available inhaled products in the market.

Drug	Quantity of API per Dose	Indication	Ref
Salbutamol	100–200 μg	Asthma	[34]
Fluticasone propionate	50–500 μg	Prophylaxis of asthma	[35]
Colistimethate sodium	80–125 mg	Treatment of pneumonia	[36]
Tiotropium	10–18 μg	Maintenance of COPD	[37]
Nedocromil sodium	2 mg	Prophylaxis of asthma	[38]
Zanamivir	5 mg	Treatment of influenza	[39]
Mannitol	5–40 mg	Treatment of cystic fibrosis as add-on therapy to standard care	[40]
Budesonide with formoterol	100–400 μg with 4.5–12 μg	Maintenance of asthma	[41]
Ciclesonide	80–160 μg	Prophylaxis of asthma	[42]

**Table 3 pharmaceutics-13-01056-t003:** Location of infections causative microorganisms within the respiratory tract.

Organism	Type	Location/Generation	Reference
**Bacteria**	*Pseudomonas aeruginosa*	Non-mucoid strain located mainly in the conduction airways Mucoid strain present throughout the respiratory zone	[47]
	*Staphylococcus aureus*	Nasal cavity Generation 0	[48]
	*Mycobacterium tuberculosis* *Chlamydia pneumonia*	Alveolar surfaces Macrophages in lungs Generation 20–22 Generation 21–23 Alveolar type 2 cells	[4] [2]
**Fungi**	*Aspergillus* spp.	Terminal bronchioles, Terminal airways Generation 16–23	[49]
**Viruses**	Herpes Simplex Virus	Oropharynx–Generation 0	[48]

**Table 5 pharmaceutics-13-01056-t005:** Licensed inhaled antibacterials currently available on the market.

Antibacterial Drug	Form	Available Strength	Ref
Tobramycin	Nebuliser liquid	300 mg/5 mL, 300 mg/4 mL, 170 mg/1.7 mL	[78]
	Inhalation powder	28 mg (1 dose = 4 × 28 mg inhalations)	
Colistimethate sodium	Inhalation powder	1,662,500 IU ≈ 125 mg	[79,80]
	Powder for nebuliser solution	1,000,000 IU ≈ 80 mg	
Aztreonam	Powder and solvent for nebuliser solution	75 mg	[81]

**Table 6 pharmaceutics-13-01056-t006:** Carrier free DPIs and the technology used to produce the powders.

API	Dose in Capsule	Conditions Used	Ref
Indomethacin	3 mg	Spray drying an aqueous-based feed to form microparticles	[89]
Ethionamide + moxifloxacin	20 mg	Spray drying using a mini spray dryer	[90]
Ketotifen	20 mg	Spray drying with different solvents (water, ethanol and water-ethanol mix)	[91]
Sildenafil	5 mg	Spray drying using a mini spray dryer	[92]
Ibuprofen	50 mg	Air-jet milling to produce micronised samples	[93]
Sodium cromoglycate	20 mg	Pelletised	[94]
Tobramycin + clarithromycin	22.72 mg tobramycin, 2.27 mg clarithromycin	Spray drying	[95]
Salbutamol sulphate	5.1–7.1 mg	Gas-phase coating method to produce L-leucine coated powders	[96]
Tranexamic acid	38 mg	Spray drying	[97]
Roflumilast	20 mg	Spray drying with hydroxypropyl-β-cyclodextrin	[98]
Tobramycin	28 mg	Micronised using a Labomill jet milling system	[99]
Meloxicam potassium	1.3 mg	Cospray drying	[100]
Ciprofloxacin + colistin	10 mg	Cospray drying	[101]
Netilmicin	30 mg	Cospray drying	[102]

**Table 7 pharmaceutics-13-01056-t007:** Application of coamorphous solid dispersions prepared using different preparation methods. The inclusion of PASD of ciprofloxacin was shown to identify the apparent benefits of polymeric solid dispersions compared to coamorphous solid dispersions.

API	Prime Excipient	Preparation Method	Fine Particle Fraction (FPF) (%)	Ref
Cyclosporin A (CsA)	Lactose, methylcellulose and erythritol	Jet-milling and freeze drying	54	[139]
Ciprofloxacin	Polyvinyl alcohol PVA	Spray drying	25 ± 2.1 after 6 months	[121]
Ciprofloxacin	No excipient	Spray drying	67.35 ± 1.1 after 6 months	[121]
Ciprofloxacin	Leucine	Spray drying	79.78 ± 1.2 after 6 months	[121]
Ciprofloxacin	Hydroxypropyl-beta-cyclodextrin	Spray drying	36.32 ± 1.3 after 6 months	[121]
Colistin	No excipient	Spray drying	43.8 ± 4.6%	[140]
Colistin	No excipient	Jet milling	28.4 ± 6.7 %	[140]
Colistin	L-leucine	Spray drying	43.8 ± 4.6% (no difference from Spray dried alone)	[140]
Ciprofloxacin	No excipient	Spray drying	28.0 ± 3.2%	[141]
Ciprofloxacin	Lactose, sucrose, trehalose, L-leucine	Spray drying	Lactose (43.5 ± 3.3%), sucrose (44.0 ± 4.3%), trehalose (44.0 ± 1.9%), L-leucine (73.5 ± 7.1%)	[141]
Thymopentin	Lactose/mannitol, Leucine, poloxamer 188	Spray drying	44.8%, 45.6%, 44.9%, 43.8%	[142]
CsA	Inulin	Spray freeze drying	>50	[143]
Tacrolimus	Mannitol	Thin-film freeze-drying	83.3	[144]
Tacrolimus	Raffinose	Thin-film freeze-drying	69.2	[145]
Tacrolimus	Lactose	Thin-film freeze-drying	68.7	[145]

**Table 8 pharmaceutics-13-01056-t008:** Cocrystalline APIs, their coformers, technology used and molecular weights.

API	Coformer	Technology Used	Molecular Weight (g/mol)		
			API	Coformer	Ref
Itraconazole	Succinic acid	Jet-milling	705.63	118.09	[163]
	L-tartaric acid			150.09	
Levofloxacin	Metacetamol	Grinding and heating	361.37	151.16	[178]
Pyrazinamide	3-Hydroxy benzoic acid	Slow evaporation and neat grinding	123.11	138.12	[179]
Dapsone	Caffeine	Slow evaporation, liquid-assisted grinding, spray drying	248.30	194.19	[180]
Nitrofurantoin	Melamine	Slow evaporation	238.16	126.12	[181]
Telaprevir	4-aminosalicylic acid	Ball milling	679.85	153.14	[182]
Trimethoprim	Glutarimide	Slow evaporation	290.32	113.11	[183]
Telmisartan	Gentisic acid	Slurry approach	514.62	154.12	[184]
	Maleic acid			116.07	
Sulfadimidine	4-aminosalicylic acid	Liquid-assisted comilling	278.33	153.14	[185]
Isoniazid	Ferulic acid	Liquid-assisted grinding	137.14	194.18	[186]
	Caffeic acid			180.16	
	Vanillic acid			168.15	
Adefovir	Stearic acid	Antisolvent precipitation	273.19	284.50	[187]
Acyclovir	Fumaric acid	Liquid-assisted grinding	225.20	116.07	[188]
Meloxicam	Aspirin	Solution, slurry and solvent drop methods	351.40	180.16	[189]
Theophylline	Oxalic acid	Spray freeze drying	180.16	90.03	[174]
Niclosamide	Nicotinamide	Spray drying	327.12	122.12	[190]
Lomefloxacin	Barbituric acid	Slow evaporation	351.35	128.09	[191]
	Isophthalic acid			166.13	
Enoxacin	Oxalic acid	Slow evaporation	320.32	90.03	[192]
	Malonic acid			104.06	
	Fumaric acid			116.07	
Sulfaguanidine	Thiobarbutaric acid	Slow evaporation	214.24	144.15	[193]
	1,10-phenanthroline			180.20	

## Data Availability

Not applicable.

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
