# Peer review of "Pulmonary Drug Delivery of Antimicrobials and Anticancer Drugs Using Solid Dispersions"

_pharmaceutics, 2021, doi:10.3390/pharmaceutics13071056_

Round 1
Reviewer 1 Report
I found the review interesting and useful for readers. It is well organized and includes a lot of information. In general, the paper is well referenced, but a few references which are missing would be very relevant to add.
I mean, in particular:
- Ogienko, A. G. et al. (2017). Large porous particles for respiratory drug delivery. Glycine-based formulations. European Journal of Pharmaceutical Sciences, 110, 148-156.
- Chvatal, A. et al. (2019). Formulation and comparison of spray dried non-porous and large porous particles containing meloxicam for pulmonary drug delivery. International journal of pharmaceutics, 559, 68-75.
- Biddiscombe, M. F., & Usmani, O. S. (2018). Is there room for further innovation in inhaled therapy for airways disease?. Breathe, 14(3), 216-224.
- Ogienko, A. G. et al, (2018). Cryosynthesis of Co-Crystals of Poorly Water-Soluble Pharmaceutical Compounds and Their Solid Dispersions with Polymers. The “Meloxicam–Succinic Acid” System as a Case Study. Crystal Growth & Design, 18(12), 7401-7409.
- Adali, M. B., et al (2020). Spray freeze-drying as a solution to continuous manufacturing of pharmaceutical products in bulk. Processes, 8(6), 709.
- Indermun, S., et al (2020). Porous particulate platforms for enhanced pulmonary delivery of bioactives. In Targeting Chronic Inflammatory Lung Diseases Using Advanced Drug Delivery Systems (pp. 359-373). Academic Press.
- Zhu, C. et al (2020). Inhalable Nanocomposite Microparticles with Enhanced Dissolution and Superior Aerosol Performance. Molecular Pharmaceutics, 17(9), 3270-3280.
- Chaurasiya, B., & Zhao, Y. Y. (2021). Dry Powder for Pulmonary Delivery: A Comprehensive Review. Pharmaceutics, 13(1), 31.
Author Response
We thank the reviewer for their valuable feedback, we have addressed the comment by including suggested references. The added references were incorporated and highlighted using track changes. We have also revised other parts of the review to include additional relevant references.
Reviewer 2 Report
Overall, the manuscript is well-written and comprehensive. I think the readers from pharmaceutics will benefit from this review. There are only some comments to be addressed before publication:
- The last section of the manuscript should be filled such as author contribution, funding, consent statement....
- I think it should be added a section regarding the use of nanomedicines through the lung. For example, nano in microparticles to keep the advantages of the nanosize. See this review: "Nebulised antibiotherapy: conventional versus nanotechnology-based approaches, is targeting at a nano scale a difficult subject?"
- It has not been mentioned the use of forming complexes with different cations to enhance the retention time of the API. This is specially used in the delivery of antimicrobials and anticancer drugs. This has been widely explored with ciprofloxacine"New aerosol formulation to control ciprofloxacin pulmonary concentration"
- I think it would be useful to add a section on new patents on this topic.
Author Response
We thank the reviewer for their valuable feedback.
- The last section of the manuscript should be filled such as author contribution, funding, consent statement... **Response**We have now added this information at the end of the review article.
- I think it should be added a section regarding the use of nanomedicines through the lung. For example, nano in microparticles to keep the advantages of the nanosize. See this review: "Nebulised antibiotherapy: conventional versus nanotechnology-based approaches, is targeting at a nano scale a difficult subject?" **Response**We have now added this suggested section and was highlighted using the track changes tool.
- It has not been mentioned the use of forming complexes with different cations to enhance the retention time of the API. This is specially used in the delivery of antimicrobials and anticancer drugs. This has been widely explored with ciprofloxacine"New aerosol formulation to control ciprofloxacin pulmonary concentration". **Response**We have now added this suggested section and was highlighted using the track changes tool.
- I think it would be useful to add a section on new patents on this topic. **Response** we value the reviewer's comment however we would like to indicate that many of the technologies are kept as "knowhow" so that the purpose of this review. Hence, there is often limited data on patents that are relevant to the focus of this review.
Reviewer 3 Report
Overall Evaluation:
In this manuscript, the authors reviewed carrier-free technologies that are based on co-amorphous solid dispersions and cocrystals of delivering the antimicrobials and anticancer drugs, so as to improve flow properties and help with delivering larger doses of the drug to the lungs. The overall summary is relatively comprehensive. However, this manuscript needs further clarification and revisions. In addition, overall logic of the paper is poor. Substantial revisions needed as noted.
Specific Comments:
- In the abstract, “It is well established that current drug formulations are associated with extremely low lung deposition”. This statement is not clear. Should be reworded.
- The data of “30 to 1” that appears in the abstract is inappropriate. This data only represents the highest ratio of the carrier to drug particles, and does not represent all.
- The logicality and coherence of the introduction is poor and should be improved. For instance, line 37-59 may be more appropriate as the first paragraph of the introduction.
- In the introduction, there are some vague expressions, please check carefully. For example, the pulmonary route has been used to treat different conditions such as asthma and chronic obstructive pulmonary disease (COPD). Does “different conditions” refer to diseases related to lungs only?
- “when antimicrobials are delivered directly to the lungs, higher doses ensure sufficient lung concentration for tackling the infection”. However, higher doses of antibiotics can lead to its substantial residue in the body.
- The section of “3. The Necessity to Deliver Larger Doses to Treat Respiratory Tract Conditions” does not correspond to the article title. “Lung Infections and Cancer” may be more suitable instead of “Respiratory Tract Conditions”.
- Line 133-135 “Many of the higher inhalable doses are administered via the use of nebulisers, which may require guidance and can also be complicated for patients to use”. A reference or evidence is missing.
- The authors state that “The nature of lung cancer suggests that it is more effectively treated by direct delivery to the lungs”. What is the nature of lung cancer?
- It is interesting to know the type of carrier particles other than alpha-lactose monohydrate, the ratio of carrier to drug particles, and the dose delivered to the lungs of antimicrobials and anticancer drugs, but this paper is missing in this research.
- The sentence “FPF refers to the amount of particles smaller than 5 μm in size in each actuation, divided by the total emitted dose of the API released in each actuation available to be inhaled [18]. DPIs that do not contain carrier particles therefore have greater potential for delivering higher doses to the lung” is difficult to understand.
- “However, the crystalline state often shows poor solubility, due to the high energy required break this structure which creates a major problem for bringing APIs to the market”. A reference or evidence is missing.
- The authors simple introduced that CASDs can be obtained through different preparation methods such as quench cooling, milling and solvent evaporation methods. The advantages and disadvantages of the different preparation methods and their requirements for antimicrobials and anticancer drugs should be further introduced.
Author Response
In this manuscript, the authors reviewed carrier-free technologies that are based on co-amorphous solid dispersions and cocrystals of delivering the antimicrobials and anticancer drugs, so as to improve flow properties and help with delivering larger doses of the drug to the lungs. The overall summary is relatively comprehensive. However, this manuscript needs further clarification and revisions. In addition, overall logic of the paper is poor. Substantial revisions needed as noted.
**Response**we thank the reviewer for their valuable comment and we have made changes to address their concerns and highlighted them using the "track changes" tool.
- In the abstract, “It is well established that current drug formulations are associated with extremely low lung deposition”. This statement is not clear. Should be reworded. **Response**This statement has been changed to "It is well established that currently available inhaled drug formulations are associated with extremely low lung deposition."
- The data of “30 to 1” that appears in the abstract is inappropriate. This data only represents the highest ratio of the carrier to drug particles, and does not represent all. **Response**This statement has been changed to "While this seems like a practical approach, in some formulations the ratio between the carrier to drug particles can be as much as 30 to 1."
- The logicality and coherence of the introduction is poor and should be improved. For instance, line 37-59 may be more appropriate as the first paragraph of the introduction. **Response** we have moved the paragraph as suggested by the reviewer
- In the introduction, there are some vague expressions, please check carefully. For example, the pulmonary route has been used to treat different conditions such as asthma and chronic obstructive pulmonary disease (COPD). Does “different conditions” refer to diseases related to lungs only? **Response** We thank the reviewer for their comment, we have now added the word "lung conditions to indicate that these conditions are often local to the lungs. We hope this addition clarifies the purpose of this sentence.
- “when antimicrobials are delivered directly to the lungs, higher doses ensure sufficient lung concentration for tackling the infection”. However, higher doses of antibiotics can lead to its substantial residue in the body. **Response** We thank the reviewer for their comment, we have replaced the word "sufficient" with "optimum" just to indicate that the level of antimicrobials can be adjusted. As indicated elsewhere in the review, the amount of the drug reaching the lungs is low hence increasing the dose will improve the efficacy of reaching an optimum dose.
- The section of “3. The Necessity to Deliver Larger Doses to Treat Respiratory Tract Conditions” does not correspond to the article title. “Lung Infections and Cancer” may be more suitable instead of “Respiratory Tract Conditions”. **Response** changed as per reviewer's suggestion
- Line 133-135 “Many of the higher inhalable doses are administered via the use of nebulisers, which may require guidance and can also be complicated for patients to use”. A reference or evidence is missing. **Response** Reference has been added as per reviewer's suggestion
- The authors state that “The nature of lung cancer suggests that it is more effectively treated by direct delivery to the lungs”. What is the nature of lung cancer? **Response** We further clarified this sentence by adding supportive work from the literature "Sardeli et al. suggested the need to use inhaled immunotherapy as opposed to intravenous administration so that to avoid systemic side effects and achieve a localized effect [52]."
- It is interesting to know the type of carrier particles other than alpha-lactose monohydrate, the ratio of carrier to drug particles, and the dose delivered to the lungs of antimicrobials and anticancer drugs, but this paper is missing in this research. **Response** We value the reviewer's comment. The focus of this review is on solid dispersions so we avoided details on the carrier particles. The point raised by the reviewer is an interesting point and we will use it as basis for our next meta-analysis.
- The sentence “FPF refers to the amount of particles smaller than 5 μm in size in each actuation, divided by the total emitted dose of the API released in each actuation available to be inhaled [18]. DPIs that do not contain carrier particles therefore have greater potential for delivering higher doses to the lung” is difficult to understand. **Response** We have reworded the sentence as per the reviewer's comment "FPF refers to the fraction of particles with a size smaller than the respirable-size divided by the total emitted dose of the inhaled API [20]."
- “However, the crystalline state often shows poor solubility, due to the high energy required break this structure which creates a major problem for bringing APIs to the market”. A reference or evidence is missing. **Response** We have added supportive references as suggested by the reviewer
- The authors simple introduced that CASDs can be obtained through different preparation methods such as quench cooling, milling and solvent evaporation methods. The advantages and disadvantages of the different preparation methods and their requirements for antimicrobials and anticancer drugs should be further introduced. **Response** We value the reviewer's comment. However, these methods are generic methods and are not specific to prepare anticancer drugs or antibiotics. The aim is to list these methods and hopefully attract readers' attention interested in using those methods to carry on research that involves these classes of drugs.
Round 2
Reviewer 3 Report
The authors have addressed the comments from the reviewers.